# Physical limits to magnetogenetics

Markus Meister*

Division of Biology and Biological Engineering, California Institute of Technology, Pasadena, United States

**Abstract** This is an analysis of how magnetic fields affect biological molecules and cells. It was prompted by a series of prominent reports regarding magnetism in biological systems. The first claims to have identified a protein complex that acts like a compass needle to guide magnetic orientation in animals (*Qin et al., 2016*). Two other articles report magnetic control of membrane conductance by attaching ferritin to an ion channel protein and then tugging the ferritin or heating it with a magnetic field (*Stanley et al., 2015*; *Wheeler et al., 2016*). Here I argue that these claims conflict with basic laws of physics. The discrepancies are large: from 5 to 10 log units. If the reported phenomena do in fact occur, they must have causes entirely different from the ones proposed by the authors. The paramagnetic nature of protein complexes is found to seriously limit their utility for engineering magnetically sensitive cells.

## Introduction

There has been renewed interest recently in the effects of magnetic fields on biological cells. On the one hand we have the old puzzle of magnetosensation: How do organisms sense the Earth's magnetic field for the purpose of navigation? The biophysical basis for this ability is for the most part unresolved. On the other hand lies the promise of 'magnetogenetics': the dream of making neurons and other cells responsive to magnetic fields for the purpose of controlling their activity with ease. The two are closely linked, because uncovering Nature's method for magnetosensation can point the way to effectively engineering magnetogenetics.

The physical laws by which magnetic fields act on matter are taught to science students in college (*Feynman et al., 1963*). Obviously those principles impose some constraints on what biological mechanisms are plausible candidates, for both magnetosensation and magnetogenetics. A recent spate of high-profile articles has put forward audacious proposals in this domain without any attempt at such reality checks. My goal here is to offer some calculations as a supplement to those articles, which makes them appear in a rather different light. These arguments should also help in evaluating future hypotheses and in engineering new molecular tools.

## Results

### A molecular biocompass?

Generally speaking, magnetic fields interact only weakly with biological matter. The reason magnetic fields are used for whole-body medical imaging, and why they have such appeal for magnetogenetics, is that they penetrate through tissues essentially undisturbed. The other side of this coin is that evolution had to develop rather special mechanisms to sense a magnetic field at all, especially one as weak as the Earth's field.

This mechanism is well understood in just one case: that of magnetotactic bacteria (*Bazylinski and Frankel, 2004*). These organisms are found commonly in ponds, and they prefer to live in the muck at the bottom rather than in open water. When the muck gets stirred up they need to return to the bottom, and they accomplish this by following the magnetic field lines down. For

*For correspondence: meister@caltech.edu

**Competing interests:** The author declares that no competing interests exist.

**eLife digest** How biological systems interact with magnetic fields is of great interest both from a basic science perspective and for technological applications. Certain animal species can sense the Earth's magnetic field for the purposes of navigation. How that compass sense works is perhaps the last true mystery of sensory biology. If we knew how the magnetic field affects the activity of nerve cells, we could harness that mechanism to create new biomedical tools. One technological goal is to genetically engineer specific cells in the brain or elsewhere so their activity can be controlled using an external magnet. This dream has been called "magnetogenetics".

In recent months a string of reports claimed to have solved both the scientific and the technological challenges of magnetogenetics. They all involved the discovery or the engineering of protein molecules that are sensitive to magnetic fields. Markus Meister has now checked whether those claims were consistent with well-established physical laws.

For each case, Meister calculated how strongly the protein in question would link magnetic fields to cellular activity. The results show that the predicted effects are too weak to account for the reported measurements by huge margins: between five and ten orders of magnitude. It therefore appears that none of these reports have hit on a solution to magnetogenetics.

All of the proposed proteins use iron atoms to couple to the magnetic field, but Meister concludes that these proteins contain far too few iron atoms. How safe is that conclusion? There has been enormous technological interest in making tiny magnets; for example, to design the ever-denser data storage drives inside computers. Hence the magnetism of small clusters of atoms is exceedingly well understood. If any of the biological reports of magnetogenetics turned out correct, they would force a revolutionary rethinking of basic physics.

With the recognition that magnetogenetics remains unsolved, and that different approaches are needed, Meister hopes that other investigators will feel motivated to continue innovating in this area.

that purpose, the bacterium synthesizes ferrimagnetic crystals of magnetite and arranges them in a chain within the cell. This gives the bacterium a permanent magnetic moment, and allows it to act like a small compass needle. The cell's long axis aligns with the magnetic field and flagella in the back of the cell propel it along the field lines. It has been suggested that magnetosensation in animals similarly relies on a magnetite mechanism, for example by coupling the movement of a small magnetic crystal to a membrane channel (*Kirschvink et al., 2001*). A competing proposal for magnetosensation suggests that the magnetic field acts on single molecules in certain biochemical reactions (*Ritz et al., 2010*). In this so-called 'radical pair mechanism' the products of an electron transfer reaction depend on the equilibrium between singlet and triplet states of a reaction intermediate, and this equilibrium can be biased by an applied magnetic field. These two hypotheses and their respective predictions for magnetosensation have been reviewed extensively (*Johnsen and Lohmann, 2005*; *Kirschvink et al., 2010*).

On this background, a recent article by *Qin et al. (2016)* introduces a new proposal. As for magnetotactic bacteria, the principle is that of a compass needle that aligns with the magnetic field, but here the needle consists of a single macromolecule. This putative magnetic receptor protein was isolated from the fruit fly and forms a rod-shaped multimeric complex that includes 40 iron atoms. The authors imaged individual complexes by electron microscopy on a sample grid. They claim (1) that each such rod has an intrinsic magnetic moment, and (2) that this moment is large enough to align the rods with the earth's magnetic field: "about 45% of the isolated rod-like protein particles oriented with their long axis roughly parallel to the geomagnetic field". We will see that neither claim is plausible based on first principles:

## Could the protein complex have a permanent dipole moment?

The smallest iron particles known to have a permanent magnetic moment at room temperature are single-domain crystals of magnetite ($Fe_3O_4$), about 30 nm in size (*Dunlop, 1972*). Those contain about 1 million iron atoms, closely packed to produce high exchange interaction, which serves to

coordinate their individual magnetic moments (*Feynman et al., 1963*; Ch 37). The protein complex described by *Qin et al. (2016)* contains only 40 Fe atoms, and those are spread out over a generous 24 nm. There is no known mechanism by which these would form a magnetic domain and thus give the complex a permanent magnetic moment. Despite intense interest in making single-molecule magnets, their blocking temperature – above which the magnetic moment fluctuates thermally – is still below 14 degrees Kelvin (*Demir et al., 2015*). So the amount of iron in this putative molecular compass seems too small by about 5 log units.

## Could individual complexes align with the earth's field?

Let us suppose generously that the 40 Fe atoms could in fact conspire – by a mechanism unknown to science – to align their individual spins perfectly, and to make a single molecule with a permanent magnetic moment at room temperature. How well would this miniature compass needle align with the earth's magnetic field? This is a competition between the magnetic force that aligns the particle and thermal forces that randomize its orientation. What is that balance?

An atom with $n$ unpaired electrons has an effective magnetic moment of

$$\mu_{\text{eff}} = \sqrt{n(n+2)}\mu_{\text{B}}, \tag{1}$$

where (In the spirit of order-of-magnitude calculations, I will use single-digit precision for all quantities)

$$\mu_{\text{B}} = \text{Bohr magneton} = 9 \times 10^{-24} \frac{\text{J}}{\text{T}}, \tag{2}$$

For iron atoms, $n$ is at most 5, and a complex of 40 aligned Fe atoms would therefore have a magnetic moment of at best

$$m = 40 \times \sqrt{5(5+2)}\mu_{\text{B}} = 2 \times 10^{-21} \frac{\text{J}}{\text{T}}. \tag{3}$$

The interaction energy of that moment with the earth's field (about 50 μT) is at most

$$mB_{\text{Earth}} = 1 \times 10^{-25} \text{ J}. \tag{4}$$

Meanwhile the thermal energy per degree of freedom is

$$kT = 4 \times 10^{-21} \text{ J}. \tag{5}$$

The ratio between those is

$$\frac{mB_{\text{Earth}}}{kT} = 2 \times 10^{-5}. \tag{6}$$

That is the degree of alignment one would expect for the protein complex. Instead, the authors claim an alignment of 0.45. Again, this claim exceeds by about 5 log units the prediction from basic physics, even allowing for an unexplained coordination of the 40 Fe spins. Clearly the reported observations must arise from some entirely different cause, probably unrelated to magnetic fields.

## An ion channel gated by magnetic force?

With the goal of controlling the activity of neurons, *Wheeler et al. (2016)* reported the design of a molecular system intended to couple magnetic fields to ionic current across the cell membrane. Their single-component protein consists of a putative mechano-sensitive cation channel (TRPV4) fused on the intracellular face to two subunits of ferritin. The hope was that "the paramagnetic protein would enable magnetic torque to tug open the channel to depolarize cells". Indeed, the report includes experimental results from several preparations suggesting that neural activity can be modulated by static magnetic fields (There is a similar claim in *Stanley et al. (2015)*; but the evidence is scant and hard to interpret: only 18 of ~2000 cells 'responded' (their Supplementary Figure 10)). What could be the underlying biophysical mechanism?

Ferritin is a large protein complex with 24 subunits that forms a spherical shell about 12 nm in diameter. *Wheeler et al. (2016)* suppose optimistically that the two subunits of ferritin attached to

the channel protein are able to nucleate an entire 24-subunit ferritin complex. The hollow core of this particle can be filled with iron in the form of a ferric hydroxide (*Arosio et al., 2009*). At room temperature ferritin has no permanent magnetization: it is strictly paramagnetic or superparamagnetic (*Papaefthymiou, 2010*). Unlike the magnetite particles in magnetotactic bacteria, the iron core of ferritin is too small (~5 nm) to sustain a permanent dipole moment (blocking temperature ~40 K). Instead the direction of the Fe spins in the core fluctuates thermally. An external magnetic field biases these fluctuations, producing a magnetic moment $m$ proportional to the field $B$ of

$$m = \xi B, \tag{7}$$

where $\xi$ is the magnetizability of a ferritin particle. This quantity can be derived from bulk measurements of ferritin magnetic susceptibility (see Methods) at

$$\xi = 2.4 \times 10^{-22} \, \frac{\mathrm{J}}{\mathrm{T}^2}. \tag{8}$$

I will consider four scenarios by which such a ferritin particle might be manipulated with an external magnetic field. In the first, the magnetic field has a gradient, and the particle is pulled in the direction of higher field strength. In the second, the force arises from interactions among neighboring ferritins through their induced magnetic moments. In the third, the magnetic field exerts a torque assuming that the ferritin core is anisotropic, with a preferred axis of magnetization. Finally, the collective pull of many ferritins on the cell membrane may induce a stress that opens stretch-activated channels.

## A magnetic field gradient pulls on ferritin (*Figure 1a*)

Paramagnetic particles experience a force that is proportional to the magnetic field gradient and the induced magnetic moment (*Feynman et al., 1963*; Ch 35; *Pankhurst et al., 2003*). In the experiments of *Wheeler et al. (2016)* the field strength was ~0.05 T and the field gradient ~6.6 T/m (their Supplementary Figure 2). What is the resulting force on a ferritin particle?

The interaction energy between the moment and the magnetic field is

$$U = -\frac{1}{2} \, mB, \tag{9}$$

where the factor of $1/2$ arises because the moment $m$ is in turn induced by the field (*Jackson, 1998*; Ch 5.16). The force produced by the field gradient is the spatial derivative of that energy, namely

$$F_1 = -\frac{d}{dx}U = \xi B \frac{dB}{dx} = 2 \times 10^{-22} \times 0.05 \times 7 \, \mathrm{N} = 7 \times 10^{-23} \, \mathrm{N}. \tag{10}$$

This would be the force exerted by one ferritin complex on its linkage under the reported experimental conditions.

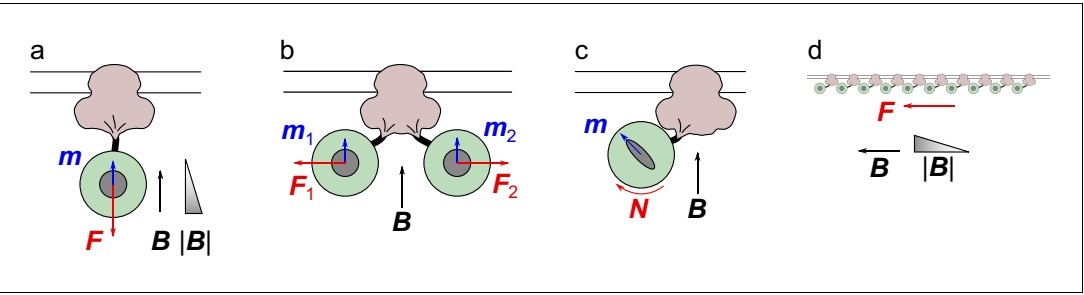

**Figure 1.** A TRPV4 channel (pink) inserted in the membrane with a ferritin complex (green) attached on the cytoplasmic side, approximately to scale. The magnetic field $B$ induces a moment $m$ in the ferritin core, leading to a force $F$ or a torque $N$ on the ferritin particle, and resulting forces tugging on the channel. See text for details.

How does this compare to the force needed to open an ion channel? That has been measured directly for the force-sensitive channels in auditory hair cells (*Howard and Hudspeth, 1988*), and amounts to about $2 \times 10^{-13}$ N. So this mechanism for pulling on ferritin seems at least 9 log units too weak to provide an explanation.

## Two ferritins pull on each other (*Figure 1b*)

As proposed by *Davila et al. (2003)*, neighboring paramagnetic particles linked to the cell membrane could tug on each other by the interaction between their magnetic moments, rather than by each being drawn into a magnetic field gradient. If the field is oriented parallel to the cell membrane, then nearby ferritins will have induced magnetic moments that are collinear and thus attract each other. If the field is perpendicular to the membrane their magnetic moments will repel (*Figure 1b*). These dipole-dipole interactions decline very rapidly with distance. For example, in the attractive configuration the force between two dipoles of equal magnetic moment $m$ at distance $d$ is given by

$$F = \frac{3\mu_0}{2\pi} \frac{m^2}{d^4},$$ (11)

where

$$\mu_0 = 4\pi \times 10^{-7} \frac{\mathrm{N}}{\mathrm{A}^2}$$ (12)

is the vacuum permeability. The strongest interaction will be between two ferritins that are nearly touching, so that $d = 2R = 12\,\mathrm{nm}$. In that situation one estimates that

$$F_2 = \frac{3\mu_0}{2\pi} \frac{(\xi B)^2}{(2R)^4} = 3 \times 10^{-21}\,\mathrm{N}.$$ (13)

Unfortunately we are again left with an exceedingly tiny force, about 8 log units weaker than the gating force of the hair cell channel.

What if the mechano-sensitive channel used in this study is simply much more sensitive to tiny forces than the channel in auditory hair cells? An absolute limit to sensitivity is given by thermal fluctuations. Whatever molecular linkage the ferritin is pulling on, it needs to provide at least $kT$ of energy to that degree of freedom to make any difference over thermal motions. Because of the steep distance dependence, the force between ferritins drops dramatically if they move just one radius apart. The free energy gained by that motion compared to the thermal energy is approximately

$$\frac{F_2 R}{kT} = \frac{3 \times 10^{-21} \times 6 \times 10^{-9}}{4 \times 10^{-21}} = 4 \times 10^{-9},$$ (14)

again 8 log units too small to have any noticeable effect.

## The magnetic field exerts a torque on the ferritin (*Figure 1c*)

Although at room temperature ferritin has no permanent magnetic moment, its induced moment may exhibit some anisotropy. In general this means that the iron core is more easily magnetized in the 'easy' direction than orthogonal to it. For example, this may result from an asymmetric shape of the core. While the exact value of that anisotropy is unknown, we can generously suppose it to be infinite, so the ferritin particle has magnetizability $\xi$ in one direction and zero in the orthogonal directions. Thus the induced magnetic moment may point at an angle relative to the field (*Figure 1c*), resulting in a torque on the ferritin particle that could tug on the linkage with the channel protein.

However, the magnitude of such effects is again dwarfed by thermal fluctuations: The interaction energy between the moment and a magnetic field pointing along the easy axis is

$$U_\parallel = -\frac{1}{2}mB = -\frac{1}{2}\xi B^2 = -3 \times 10^{-25}\,\mathrm{J}$$ (15)

and zero with the field orthogonal. This free energy difference is about 4 log units smaller than the

thermal energy. Following the same logic as for Qin et al's compass needle, the magnetic field can bias the alignment of the ferritins by only an amount of $10^{-4}$. Another way to express this is that any torque exerted by the ferritin on its ion channel linkage will be 10,000 times smaller than the thermal fluctuations in that same degree of freedom.

## Many ferritins exert a stress that gates mechanoreceptors in the membrane (*Figure 1d*)

Perhaps the magnetic responses are unrelated to the specific linkage between ferritin and a channel protein. Instead one could imagine that a large number of ferritins exert a collective tug on the cell membrane, deforming it and opening some stress-activated channels in the process. The membrane stress required to gate mechanoreceptors has been measured directly by producing a laminar water flow over the surface of a cell: For TRPV4 channels it amounts to ~20 dyne/cm$^2$ (*Soffe et al., 2015*); for Piezo1 channels ~50 dyne/cm$^2$ (*Ranade et al., 2014*). Suppose now that the membrane is decorated with ferritins attached by some linkage, and instead of viscous flow tugging on the surface one applies a magnetic field gradient to pull on those ferritins with force $F_1$ (*Equation 10*). The density of ferritins one would need to generate the required membrane stress is

$$\frac{20\ \mathrm{dyn/cm^2}}{7 \times 10^{-23}\ \mathrm{N}} = 3 \times 10^{10}\ \frac{\mathrm{ferritins}}{\mu \mathrm{m^2}} \qquad (16)$$

Unfortunately, even if the membrane is close-packed with ferritin spheres, one could fit at most $10^4$ on a square micron. So this hypothetical mechanism produces membrane stress at least 6 log units too weak to open any channels.

## An ion channel gated by magnetic heating?

For a different mode of activating membrane channels, *Stanley et al. (2015)* combined the expression of ferritin protein with that of the temperature-sensitive membrane channel TRPV1. The hope was that a high-frequency magnetic field could be used to heat the iron core of ferritin, leading to a local temperature increase sufficient to open the TRPV1 channels, allowing cations to flow into the cell. *Stanley et al. (2015)* compared three different options for interaction between the ion channels in the plasma membrane and the ferritin protein: In one case the ferritin was expressed in the cytoplasm, in another it was targeted to the membrane by a myristoyl tail, and in the third it was tethered directly to the channel protein by a camelid antibody linkage. The direct one-to-one linkage between ferritin and ion channel worked best for generating Ca influx via high-frequency magnetic fields, leading the authors to conclude that "Because temperature decays as the inverse distance from the particle surface, heat transfer is likely to be most efficient for this construct, suggesting that heat transfer from the particle could be limiting the efficiency of the other constructs." Here I consider whether heat transfer from the ferritin particle is a likely source of thermal activation for the TRPV1 channel at all.

Magnetic heating of nanoparticles is indeed a very active area of research (*Pankhurst et al., 2003*). A sample biomedical application is to inject nanoparticles into cancerous tissue, and then damage the tumor selectively by magnetic heating (*Hergt et al., 2006*; *Maier-Hauff et al., 2011*). Typical nanoparticles of interest are made of magnetite or maghemite, sometimes doped with other metals, and measure some tens of nanometers in size (*Hergt et al., 2006*). A typical heating apparatus for small preparations – like in the experiments of *Stanley et al. (2015)* – consists of an electric coil with a few windings, several centimeters in diameter, that carries a large oscillating current. The magnetic fields generated inside the coil are on the order of tens of kA/m at frequencies of several 100 kHz (The literature sometimes refers misleadingly to heating by "radio waves" or "electromagnetic radiation". At these frequencies the wavelength of a radio wave is about a kilometer, so of no practical relevance to the experiments. There is no radiation involved in the interaction between the solenoid coil and the nanoparticle).

Owing to the small size of the nanoparticles, the physics of heating are quite different from the processes in our kitchen. A microwave oven heats water primarily by flipping molecular dipoles in an oscillating electric field. And an induction stove works by inducing electric eddy currents in the pot's bottom with an oscillating magnetic field. Neither of these electric effects plays any role for nanoparticles. Instead the heat is generated purely by magnetic forces (*Hergt et al., 2006*). Part of this

comes from reorienting the magnetization of the material at high frequency, which is opposed by the particle's magnetic anisotropy, causing dissipation and heat. For larger nanoparticles, the oscillating magnetic field may also make the particle move, with resulting dissipation from external friction in the surrounding medium. These physical processes have been modeled in great detail, and there is a large experimental literature to determine the heating rates that can be accomplished with different kinds of nanoparticles. A figure of merit is the "specific loss power (SLP)", namely the heating power that can be generated per unit mass of the magnetic material (see Materials and methods). What sort of heating rate would we expect for the ferritin particles used by *Stanley et al. (2015)*?

Given the long-standing interest in ferritin for medical engineering (*Babincova et al., 2000*), the extensive research on its magnetic properties (*Papaefthymiou, 2010*), and the ease with which magnetic heating can be measured, it is surprisingly difficult to find any published evidence for magnetic heating in ferritin. One report on the subject concludes simply that there is none: ferritin shells reconstituted with a magnetite core produced no measurable magnetic heating (SLP < 0.1 W/g), whereas doping the iron with varying amounts of cobalt did produce some modest heating rates (*Fantechi et al., 2014*). Why is native ferritin such a poor heater? Both theory and experiment show that the efficiency of heating magnetic nanoparticles depends strongly on the particle size, and plummets steeply below 10 nm (*Fortin et al., 2007*; *Purushotham and Ramanujan, 2010*). Magnetite particles smaller than 8 nm are not considered useful for magnetic hyperthermia (*Fantechi et al., 2015*). The iron core of ferritin measures only 5–6 nm in diameter. Furthermore, the ferric hydroxide material in native ferritin has much lower magnetic susceptibility than magnetite (~8-fold, *Zborowski et al., 1996*).

So, based on the literature, the heating rate for ferritin is *too low to be measurable*. Obviously this casts doubt on the claims of *Stanley et al. (2015)* that they activated ion channels through heating ferritin. For the sake of keeping the argument alive, and to evaluate potential future developments, let us instead suppose that ferritin could be engineered to produce a specific heating rate of

$$P = 30 \frac{\text{W}}{\text{g of metal}} \tag{17}$$

This is the highest value obtained by filling the ferritin shell with cobalt-doped magnetite (*Fantechi et al., 2014*) and thus a generous estimate of what might be accomplished by future engineering of ferritin complexes inside cells. Assuming this specific heating rate, a single ferritin particle with 2400 iron atoms generates heat at a rate of

$$Q = 7 \times 10^{-18} \, \text{W} \tag{18}$$

This heat flux will produce a temperature gradient in the surrounding medium (*Figure 2*). As *Stanley et al. (2015)* state, the temperature indeed decays as the inverse distance $r$ from the particle (*Feynman et al., 1963*, Ch 12), namely

$$T(r) = \frac{Q}{4\pi\kappa} \frac{1}{r} \tag{19}$$

where

$$\kappa = 0.61 \frac{\text{W}}{\text{m} \cdot \text{K}} \tag{20}$$

is the thermal conductivity of water. Right at the surface of the ferritin sphere the temperature increase is highest, namely

$$T_{\text{ferritin}} = T(6 \text{ nm}) = 1.5 \times 10^{-10} \, \text{K} \tag{21}$$

This is a very tiny increase. Activation of a TRPV1 channel requires about 5 K of increase relative to body temperature (*Cao et al., 2013*). So the temperature increase expected, even from a futuristic optimized ferritin, is more than 10 orders of magnitude too small.

The assumption underlying *Equation (19)* is that thermal transport from the magnetic particle to the surrounding medium follows Fourier's Law, in which the heat flux is proportional to the

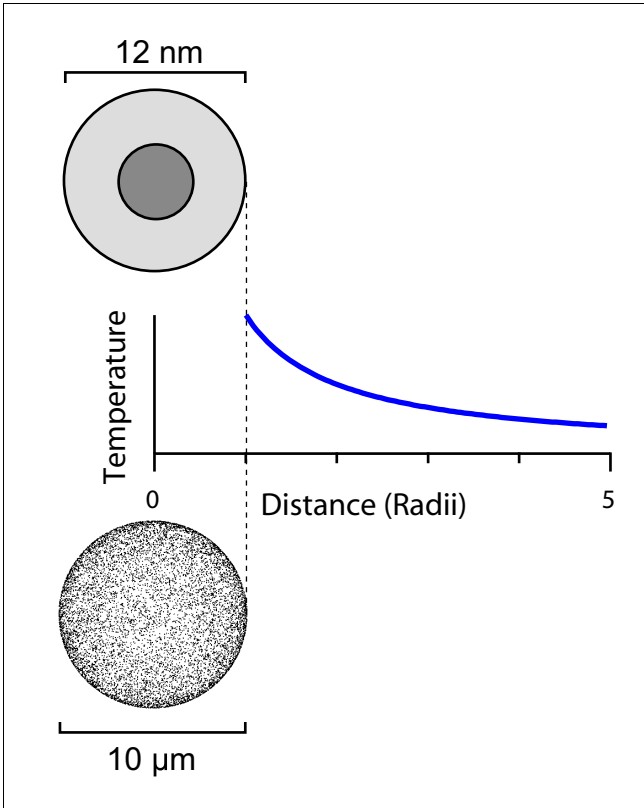

**Figure 2.** The steady-state temperature profile around a heated sphere in an infinite bath varies inversely with the distance from the center of the sphere. The same argument applies to a ferritin sphere heated from its magnetic core (top) and a spherical cell with a large number of heated ferritins on its surface (bottom).

temperature gradient. This is a good approximation, as long as the length scales of the problem are large compared to the mean free path of the heat carriers, which are phonons in the current problem. In water, the phonon mean free path is ~0.3 nm, about the size of a water molecule (*Rabin, 2002*). Indeed, all the relevant dimensions are at least 10-fold larger than that, namely the size of the ferritin particle, the size of the ion channel protein, and the distance from ferritin to ion channel. One therefore expects that non-Fourier heat transport will make only small corrections to the above results, on the order of 10% or less (*Chen, 1996*).

Another effect resulting from thermal physics at small scales is the thermal resistance to heat flow at the boundary between two materials. A given heat flux across the boundary will produce a discontinuous step in temperature between the two materials. How large is this step for heated nanoparticles? *Ge et al. (2004)* followed thermal transfer between a metal nanoparticle with organic coating and the surrounding water, and observed a thermal conductance of

$$G = 2 \times 10^8 \, \frac{\mathrm{W}}{\mathrm{m}^2 \cdot \mathrm{K}}, \tag{22}$$

largely independent of particle size. With the heat flux produced by our ferritin particle the resulting temperature step would be

$$\Delta T_{\mathrm{ferritin}} = \frac{Q}{4\pi R^2} \frac{1}{G} = 7 \times 10^{-11} \, \mathrm{K}. \tag{23}$$

Again, an exceedingly tiny contribution.

It appears that there is no possibility of raising the temperature by several degrees near a single nanoparticle, even if the heating rate were 1000-fold higher. While there have been isolated reports of such magnetic heating effects near synthetic nanoparticles (*Huang et al., 2010*; *Riedinger et al.,*

*2013*; *Piñol et al., 2015*), they have met a good amount of skepticism. As has been pointed out repeatedly, there is no known physical mechanism for such an effect (*Rabin, 2002*; *Keblinski et al., 2006*; *Gupta et al., 2010*), and it has been suggested that the underlying methods of thermometry should be reevaluated (*Dutz and Hergt, 2013*; *2014*).

Moving on from single-particle heating one may ask whether the many other ferritins expressed on the same cell, though they are at greater distance, might contribute to heating the local environment. Suppose one can express $N_{\text{ferritins}} = 10,000$ TRPV1-ferritin complexes on the surface of a spherical cell with $r_{\text{cell}} = 5\,\mu\text{m}$ radius. That is about 10-fold the natural expression level in sensory neurons. One can treat the heat production of those 10,000 ferritins as distributed evenly over the surface of the cell. Then the temperature gradient outside the cell again follows a $1/r$ profile (*Figure 2*). At the surface of the cell the resulting temperature increase will be

$$T_{\text{cell}} = \frac{Q N_{\text{ferritins}}}{4\pi\kappa r_{\text{cell}}} = 1.7 \times 10^{-9}\,\text{K} \tag{24}$$

Unfortunately this is still too low by 9 orders of magnitude. So one cannot achieve activation of single neurons this way, which is of course a central goal of genetically expressed activators.

Suppose now that one expresses this number of ferritin-TRPV1 complexes on every neuron in the brain. Would that perhaps be sufficient to heat the entire organ? At that density, the heating rate per unit mass of brain will be

$$P_{\text{brain}} = \frac{Q N_{\text{ferritins}}}{\frac{4}{3}\pi r_{\text{cell}}^3} \frac{1}{\rho_{\text{brain}}} = 1.2 \times 10^{-4}\,\frac{\text{W}}{\text{g}}, \tag{25}$$

where $\rho_{\text{brain}} = 1.03\,\text{g/cm}^3$ is the specific density. For comparison, the resting metabolic rate of brain tissue is $\sim 1.2 \times 10^{-2}\,\text{W/g}$, and the resulting heat is carried away and regulated by the processes that keep the organ's temperature stable. Heating of ferritin throughout the entire brain would therefore contribute only a 1% increase to the heat already being generated from basal activity: this will not overwhelm the homothermic regulation mechanisms sufficiently to open TRPV1 channels.

In summary, it seems very unlikely that the effects reported in *Stanley et al. (2015)* have anything to do with heating ferritin. The available evidence says that native ferritin produces no measurable magnetic heating at all. Even if we ignore that and assume a generous heating rate, namely the largest reported using a custom metal alloy for the ferritin core, the resulting effects are too small to matter by enormous factors of $10^{10}$ (single-channel activation) and $10^9$ (for single-neuron activation).

## Discussion

The calculations presented here evaluate the mechanisms that might underlie recent observations on a molecular compass (*Qin et al., 2016*) and neural activation with static magnetic fields (*Wheeler et al., 2016*) or high-frequency magnetic fields (*Stanley et al., 2015*). These calculations show that none of the biophysical schemes proposed in these articles is even remotely plausible, and a few additional proposals were eliminated along the way. The forces or torques or temperatures they produce are too small by many orders of magnitude for the desired effects on molecular orientation or on membrane channels. If the phenomena occurred as described, they must rely on some entirely different mechanism. Barring dramatic new discoveries about the structure of biological matter, the proposed routes to magnetogenetics, based on either pulling or heating a ferritin/channel complex with magnetic fields, have no chance of success.

One does have to ask why none of these authors attempted a back-of-the-envelope estimate to bolster the claims in their papers. Neither, it seems, did the referees who reviewed the manuscripts, nor the authors of three pieces that heralded these articles (*Leibiger and Berggren, 2015*; *Lewis, 2016*; *Lohmann, 2016*). Why is it important to do so? First of all, claims that violate the known laws of physics often turn out to be wrong (*Maddox et al., 1988*). There is, of course, always a small chance of discovering new physics, but only if one understands what the old physics predicts and recognizes the discrepancy. If any of the claims in these articles were substantiated – a room-temperature molecular magnet or measurable forces and heating from ferritin – their implications for our basic understanding of nanoscale matter would far outweigh their biological significance.

More importantly though, calculations are most useful when done ahead of time, to guide the design of experiments. For example, *Stanley et al. (2015)* and *Wheeler et al. (2016)* evoke an image in which the magnetic field pulls on the ferritin particles. This is possible only if the magnetic field has a strong gradient (*Figure 1a*, *Equation 10*). None of their experiments on animals were designed to produce a strong gradient, nor do the articles report what it was. It is in fact possible to pull on cells that express lots of ferritin, and this has been exploited for magnetic separation (*Owen and Lindsay, 1983*). It requires very high magnetic fields, and separation columns with a meshwork of fine steel fibers that produce strong gradients on a microscopic scale. Inserting such a wire mesh into the brain would of course negate the goal of non-invasive control.

Two other hypothetical mechanisms for the ferritin effects require a strong field but no gradient. This would be of great experimental value, because a homogeneous magnetic field could then deliver the same control signal throughout an extended volume, like the brain of a mouse. Among these, the dipole interaction between ferritins (*Figure 1b*, *Equation 14*) offers little hope. Even with a 100-fold larger field (5 T), these forces are still 4 log units too small to open a channel. That field strength represents a practical limit: Small movements of the animal, or switching of the field, will cause inductive eddy currents that activate the brain non-specifically, a phenomenon experienced also by MRI subjects (*Schenck et al., 1992*).

On the other hand, exploiting anisotropy of the ferritin particle (*Figure 1c*, *Equation 15*) may be within range of utility. A 100-fold larger field could produce torque comparable with the thermal energy, which when applied to thousands of channels might have a noticeable effect on membrane currents. To enhance the shape anisotropy of the magnetic particles, perhaps one could engineer the ferritin shell into an elongated shape. More fundamentally, it is clear that the weak effects computed here are a consequence of ferritin's paramagnetism. A particle with a permanent magnetic moment, such as the magnetosomes made by bacteria (*Bazylinski and Frankel, 2004*), could exert much larger forces, torques, and temperatures (*Hergt et al., 2006*), and may offer a physically realistic route to magnetogenetics.

With an eye towards such future developments, it is unfortunate that these three questionable claims were published, especially in high-profile journals, because that discourages further innovation. Now that the prize for magnetogenetics has seemingly been taken, what motivates a young scientist to focus on solving the problem for real? There is an important function here for post-publication peer review: It can make up for pre-publication failures and thus reopen the claimed intellectual space for future pioneers.

## Materials and methods

### Magnetizability of native ferritin

Central to the arguments about magnetogenetics is the proportionality factor $\xi$ between the magnetic moment $m$ of a single ferritin molecule and the magnetic field $B$,

$$m = \xi B. \tag{26}$$

Experimental measurements are usually performed on bulk samples of ferritin and report the magnetic susceptibility $\chi$, defined by

$$M = \chi H = \chi B / \mu_0, \tag{27}$$

where $M$ is the magnetization of the material, namely the magnetic moment per unit volume, and

$$\mu_0 = 4\pi \times 10^{-7} \frac{N}{A^2} \tag{28}$$

is the vacuum permeability. Therefore

$$\xi = \frac{\chi}{\rho \mu_0}, \tag{29}$$

where $\rho$ is the number of ferritin particles per unit volume. In practice, we will see that the reported

**Table 1.** Published measurements of specific loss power (SLP) for various magnetic particles of diameter *d*, taken at a magnetic field strength *H* and frequency *f*. The values in the column "SLP corr" are corrected for the field and frequency used by *Stanley et al. (2015)*.

| Reference | Material | d [nm] | H [kA/m] | f [kHz] | SLP [W/g] | SLP corr [W/g] | Notes |
|---|---|---|---|---|---|---|---|
| *Fortin et al. (2007)* | $Fe_2O_3$ | 5.3 | 24.8 | 700 | 4 | 2.8 | |
| *Fortin et al. (2007)* | $Fe_2O_3$ | 6.7 | 24.8 | 700 | 14 | 10 | |
| *Fortin et al. (2007)* | $Fe_2O_3$ | 8 | 24.8 | 700 | 37 | 26 | |
| *Fantechi et al. (2015)* | $Fe_3O_4$ | 8 | 12 | 183 | 6.5 | 75 | |
| *Hergt et al. (2004)* | $Fe_2O_3$ | 7 | 15 | 410 | 15 | 49 | |
| *Fantechi et al. (2014)* | ferritin with $Fe_3O_4$ | 6 | 12.4 | 183 | <0.01 | <0.1 | per mass of only the metal ions |
| *Fantechi et al. (2014)* | ferritin with $Co_{0.15}Fe_{2.85}O_4$ | 6.8 | 12.4 | 183 | 2.81 | 30 | per mass of only the metal ions |

measurements of magnetization are more often normalized by the iron content of the sample or by the mass, rather than by volume. Then the choice of $\rho$ must be adjusted accordingly.

- *Michaelis et al. (1943)* report the susceptibility $\chi_{Fe}$ of ferritin at $5.9 \times 10^{-3}$ CGS units per mole of iron in the preparation. Therefore we must divide by the number of ferritins per mole of iron, $\rho_{Fe}$. The authors report iron loading of maximally 23% w/w, which amounts to 2400 Fe atoms per ferritin, and so

$$\rho_{Fe} = \frac{N_A}{2400}, \tag{30}$$

where $N_A$ is Avogadro's number. Furthermore, note that one CGS unit of molar susceptibility is equivalent to $4\pi \times 10^{-6}$ SI units. Therefore the magnetizability of one ferritin is

$$\xi_{Mic} = \frac{\chi_{Fe}}{\rho_{Fe}\mu_0} = \frac{2400 \times 5.9 \times 10^{-3} \times 4\pi \times 10^{-6}}{6.02 \times 10^{23} \times 4\pi \times 10^{-7}} \frac{J}{T^2} = 2.35 \times 10^{-22} \frac{J}{T^2}. \tag{31}$$

As a sanity check for all the conversions, we can use the authors' statement that the susceptibility followed the Curie Law with an equivalent moment per iron atom of

$$\mu_{eff} = 3.78 \, \mu_B. \tag{32}$$

From this one derives

$$\xi_{Mic} = \frac{N\mu_{eff}^2}{3kT} = \frac{2400 \times (3.78 \times 9.27 \times 10^{-24})^2}{3 \times 4.11 \times 10^{-21}} \frac{J}{T^2} = 2.37 \times 10^{-22} \frac{J}{T^2} \tag{33}$$

in close agreement with *Equation 31*.

- *Schoffa et al. (1965)* again report the susceptibility $\chi_{Fe}$ referred to the iron content with a value of $6.05 \times 10^{-3}$ CGS units per mole of iron. Assuming again an iron loading of 2400 Fe per ferritin, this results in

$$\xi_{Sch} = 2.41 \times 10^{-22} \frac{J}{T^2}. \tag{34}$$

- *Jandacka et al. (2015)* report a susceptibility per unit mass $\chi_{mass}$ in SI units of $2.5 \times 10^{-4} Am^2/gT$. Therefore we must divide by the number of ferritins per unit mass, $\rho_{mass}$. At 2400 Fe per particle, one ferritin weighs ~580 kD, so that

$$\rho_{mass} = \frac{N_A}{5.8 \times 10^5 \, g}, \tag{35}$$

and

$$\xi_{\mathrm{Jan}} = \frac{\chi_{\mathrm{mass}}}{\rho_{\mathrm{mass}}\mu_0} = \frac{2.5 \times 10^{-4} \times 5.8 \times 10^5}{6.02 \times 10^{23}}\frac{\mathrm{J}}{\mathrm{T}^2} = 2.41 \times 10^{-22}\frac{\mathrm{J}}{\mathrm{T}^2}. \tag{36}$$

Given that these three measurements span the better part of a century using three different instruments, the agreement is remarkable. I will use the value

$$\xi = 2.4 \times 10^{-22}\frac{\mathrm{J}}{\mathrm{T}^2}. \tag{37}$$

## Magnetic heating of nanoparticles

*Table 1* summarizes some published measurements on magnetic heating of small nanoparticles with diameter below 10 nm. The loss power per unit mass (SLP) depends on the apparatus used for heating. Over the range of conditions considered here, a good approximation is that SLP varies proportionally to the frequency of the alternating magnetic field and to the square of the field strength (*Hergt et al., 2004*). The table therefore corrects all the SLP numbers to the conditions used by *Stanley et al. (2015)*: field strength $H = B/\mu_0 = 25.5 \ \mathrm{kA/m}$ and frequency $f = 465 \ \mathrm{kHz}$.

## Acknowledgements

The author thanks Bill Bialek, Justin Bois, Stephen J Royle, and an anonymous contributor on Pub-Peer for helpful comments on an earlier version.

## Additional information

### Funding
No external funding was received for this work.

### Author contributions
MM, Did everything, Conception and design, Analysis and interpretation of data, Drafting or revising the article

### Author ORCIDs
Markus Meister, http://orcid.org/0000-0003-2136-6506

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
