## [Decision Letter]

Thank you for submitting your article "Physical limits to magnetogenetics" for consideration by *eLife*. Your article has been favorably evaluated by Eve Marder (Senior editor) and four reviewers, one of whom, David E Clapham (Reviewer #1), is a member of our Board of Reviewing Editors. The following individuals involved in review of your submission have agreed to reveal their identity: Alan P. Jasanoff (Reviewer #2) and Baron Chanda (Reviewer #4).

The reviewers have discussed the reviews with one another and the Reviewing Editor has drafted this decision to help you prepare a revised submission.

Summary:

In 'Physical limits to magnetogenetics', Meister introduces a welcome degree of caution into the growing popular field of magnetogenetics, in which Fe-containing particles are engineered into proteins, in particular ion channels. He shows through a straightforward set of calculations that the field strengths, in particular, the magnetic field gradient across small regions, is much too small to induce movement of parts of channel gating proteins. The relevant comparator is background thermal energy, or *kT*, which the particle has to overcome to produce structural rearrangements in the proteins of interest. All reviewers agreed that the work merits publications, and have made a number of relatively minor suggestions to insure accuracy.

Essential revisions:

One of the reviewers suggested several modifications of wording, definitions, and equations in order that the commentary be technically accurate.

In his manuscript “Physical limits to magnetogenetics” Prof. Markus Meister undergoes a series of back-of-the-envelope calculations that put into question physical mechanisms of magnetically- driven cell signaling proposed in a number of recent reports. First of all, I would like to applaud Prof. Meister for standing up for basic physics in light of recent articles that propose bio- magnetic mechanisms that are challenging to reconcile with the laws of magnetism. Given the recent interest in magnetoreception and magnetogenetics, I am confident that this is a timely article that should serve as a guide to the community. I do believe, however, that a number of minor improvements can be made to enhance the value of the manuscript. Below are a few suggestions for further clarification:

1) The manuscript would benefit from simple diagrams that illustrate forces and fields involved in all of the discussed scenarios. Simple figures would help our colleagues to develop conceptual understanding of magnetostatics.

2) It would be clearer if all the equations were first spelled out symbolically prior to plugging in the numbers, as it would help our colleagues to use those equations in their own future calculations.

For example: Eq. (1) μeff= n(n+2)μB= 5(5+2)μB where *n* = number of unpaired electrons

3) Subsection “2. Individual complexes align with the earth's field”: “…the complex of 20 aligned Fe atoms…” should be “40 aligned Fe atoms”, to be consistent with the first sentence of the aforementioned subsection, as well as Eq. (3).

4) Magnetization and magnetic moment should not be confused. *M* – is magnetization measured in A/m, magnetic moment m is measured in Am2 (=J/T – not a standard unit).

5) Eq. (7) is not technically correct. It should be: M = *X_v_*H, where *M* is magnetization. In case of an isolated magnetic moment in air, in SI units one can write: M=Xv Bμo, where μo=4πx10−7[NA2= T.mA= JA2.m] is permeability of free space.

6) The calculation of the susceptibility in the Methods section is rather confusing. When discussing the susceptibility, the author refers to volumetric susceptibility (*X_v_*– a unitless quantity), while what he appears to mean is molar susceptibility (*X_m_* in m^3^/mol (SI)). There are some typos in text and ambiguity in equations associated with this.

Unit conversion for volumetric susceptibility is as follows: XSIv=4πXCGSvUnit conversion for molar susceptibility is: XSIm[m3mol]=4π x 10−6 XmCGS[cm3mol] (not vice versa as stated in Methods section.)

If the author had indeed used volumetric (unitless) susceptibility the units would not work out and a factor of volume would emerge as expected for the distinction between magnetization and magnetic moment.

Michaelis et al. measure ferritin molar susceptibility in CGS units as:XmCGS=5.9 x  10−3[cm3mol]

The susceptibility per Fe atom in SI units is then:

XSIFe= XmSINA= 4π x 10−6 XmCGSNA= 4π x 10−6 x 5.9 x10−3 6.02 x 1023 ≈1.2 x 10−31[m3atom]Where N_A_ is the Avodagros number.

If ferritin included 2400 Fe atoms, the moment per complex as a function of applied field B (paramagnetic material does not have an inherent moment in the absence of applied field) in free space would be:mmic= NFe⋅XSIFe⋅Bμ0=2400 ⋅XSIFeμ0 B= XSIMicμ0 B

In calculations of magnetic moment according to m= Xm Bμ0, the author seems to inherently include dividing susceptibility by the permeability of free space into the definition of susceptibility, leading to unusual units:

XSIMicμ0= 2400x1.2x10−31[m3]4π x 10−7[JA2m]= 2.9x10−28[m3]4π x 10−7[JA2m]=2.3 x 10−22 [A2m4J=JA2m4J21T2=JT2]7) Introduction, second paragraph: Feynman’s Lectures on Physics is an excellent physics reference, but the citation itself may require some editing. Profs. Leighton and Sands helped compile Feynman’s lectures into a written volume and are typically also credited with authorship, and since Feynman sadly passed away in 1988, a 2011 date without mention of the edition number may confuse some readers.

8) Subsection “1. The protein complex has a permanent dipole moment”: “Those contain about 1 million iron atoms, closely packed to produce high exchange interaction, which serves to lock their individual magnetic moments in parallel (Feynman, 2011).”

This statement could use some clarification. Magnetite is ferrimagnetic, so the moments are not all parallel, and the ordering arises from superexchange mediated by oxygen. This does not subtract from point being made about scale and blocking temperature, but the author should carefully evaluate all instances of the term “ferromagnetic.”

9) Subsection “A magneto-sensitive channel?”: “The hollow core of this particle can be filled with iron oxide (Arosio et al., 2009)” Upon examining this reference, it seems that ferritin may not store iron directly as iron oxide, but rather: “The main in vitro reaction of any ferritin type is to react with Fe(II) and induce its oxidation and deposition inside the cavity in a ferric oxyhydroxide core which is structurally similar to the mineral ferrihydrite.” (p. 590, Arosio et al.) Michaelis et al. (1943), as well as Papaefthymiou (2010) seem to generally agree on this point as well.

10) The statement that “At room temperature, ferritin is strictly paramagnetic” seems to be supported by Michaelis et al. (1943) and other literature. However, the justification that follows is somewhat unclear. A “strict” paramagnet should not exhibit magnetic ordering at any scale without an applied field. In contrast, superparamagnetic particles can retain their local magnetic order above their blocking temperature, while the direction of magnetization undergoes rapid thermal fluctuation. A Langevin curve is observed for bulk measurements of such materials, but low field susceptibility is large (“super”) compared to strict paramagnets because the magnetic moments responding to the external field are coming from comparatively large, magnetically ordered clusters. As noted in the Methods section of this manuscript, consistent measurements have been performed on biologically derived ferritin that seem to indicate paramagnetism at room temperature. However, for perspective, it may be useful to direct readers to some work that discusses the possibility of ferrimagnetic ferrihydrite phases such as Michel et al. (2010). PNAS: 107 (7) 2787-2792 doi: 10.1073/pnas.0910170107

11) Eq. (8) does not express susceptibility in units that correspond to susceptibility. Rather, it is an estimate of the moment of a ferritin when multiplied by B. The following correction should be considered (see comment 6 above for the Methods section):m= NFe⋅XSIFe⋅Bμ0= (2.4 x 10−22JT2)⋅B

Where N_Fe_ is the number of Fe per ferritin.

12) Eq. (9) does not seem correct if *M* actually represents a magnetic moment m⇀. See Feynman, Vol 2, equation 15.4. The energy of a magnetized dipole in a field is:U= −m→⋅B→= − NFe⋅XSIFe⋅B2μ0

The justification for the factor of ½ should be clarified if retained.

13) Suggested correction for Eq. (10):F1= − ddx U=2 NFe⋅XSIFe⋅Bμ0⋅dBdx

The numerical estimate should come out to be the same, but the physical reasoning is clearer, at least to this reviewer. Nonstandard representations of susceptibility should be avoided if possible.

14) In Eq. (11), magnetization and moment are again confused. Simply substitute m for M and the same or similar numerical result should be obtained. Is the author using the same assumption of applied field B as in the previous example? Two paramagnetic bodies should not exhibit attraction without an applied field.

15) For Eq. (15), see the comment on Eq. (9), paying attention to sign. The energy of the particle in the field aligned with its easy axis should be negative, so that it is less than that of the particle with B applied along the hard axis. Otherwise the torque will be in an unphysical direction.

Another important point, but only strongly encouraged, not required:

"For the sake of completeness, it might also be a good idea to add a subsection on the physicality of the RF heating approach proposed by the Stanley et al. team in their 2015 paper and a more recent 2016 paper (Nature 531: 647-50). This is the one biophysical mechanism in the magnetogenetics papers that is not yet treated in the current manuscript. At the least, it would be worth commenting briefly on the validity of the RF heating approach in the Discussion section."

[Editors' note: further revisions were requested prior to acceptance, as described below.]

Thank you for resubmitting your work entitled "Physical limits to magnetogenetics" for further consideration at *eLife*. Your revised article has been favorably evaluated by Eve Marder (Senior Editor), a Reviewing editor, and one of the original reviewers.

The manuscript has been improved but there are some remaining issues that need to be addressed before acceptance, as outlined below from Reviewer 3:

"The revised version of the manuscript ‘Physical limits to magnetogenetics’ addresses my comments to the earlier version. The added discussion of heating in magnetic nanomaterials, however, would benefit from substantial clarification, particularly in light of recent experimental observations of unexpected nanoscale heating effects. My specific suggestions are outlined below:

1) Magnetic nanoparticle (MNP) heating with low-radio frequency alternating magnetic fields has been an active area of research for 5 decades, and a number of comprehensive reviews are dedicated to the subject. Citing one of these reviews (e.g. Pankhurst, J. Phys. D Appl. Phys. 2003 or 2009) would be useful for the community. Most applications of MNP heating are in the field of cancer hyperthermia, and in fact these materials have been used in clinical trials to extend life span of glioblastoma patients by triggering tumor cell necrosis through bulk heating (Maier-Hauff, J. Neurooncol. 2011). Many of these nanomaterials are indeed based on synthetic magnetite, and its transition-metal-doped derivatives with dimensions in the range 6-60 nm. Citations to cancer hyperthermia reviews/ clinical trials should be mentioned in addition to those chosen by the author.

2) Specific loss power (SLP) is indeed the main figure of merit used to compare MNP heating efficacy, however, the reference to Hergt should be updated because linear-response theory (LRT) has a limited range of validity. It approximates the Neél relaxation time as independent of the applied field magnitude, an approximation that is most valid for MNPs with coercive field significantly exceeding the amplitude of AMF. This is not the case for many MNPs in 10s kA/m fields. Recent articles by Carrey, Mamiya and colleagues (Carrey, J. Appl. Phys. 2011; Mamiya, Sci. Rep. 2011) offer a dynamic hysteresis theory (DHT) for heat dissipation by MNPs in AMFs. According to DHT, the particle loss per cycle is predicted to saturate at large fields rather than growing without bound as LRT suggests. For quantitative experimental evidence, see (Mehdaoui, Adv. Funct. Mater. 2011; Christiansen et al. Appl. Phys. Lett. 2014). Achieving non-negligible SLPs with small particles is sometimes possible by simply increasing the AMF frequency. This exact principle has been applied by Pralle and colleagues, who used AMF with an amplitude 2 kA/m and frequency 40 MHz to heat 6 nm MnFe2O4 MNPs and actuate TRPV1 (Huang, Nat. Nanotechnol. 2010). Prolonged operation at such frequencies, however, is not advisable due to the inductive heating of the biological tissue observed for field-frequency products in excess of 5×109 kA m-1 s^-1^ (Hergt and Dutz, J. Magn. Magn. Mater. 2007). Bearing this constraint in mind, larger particles with wider hysteresis loop areas are frequently chosen as more effective at reasonable field-frequency conditions.

3) When talking about heat dissipation, the author also mentions Brown relaxation in addition to Neél relaxation (described by DHT) in passing. Brown relaxation, i.e. physical rotation of the particles, is only applicable to MNPs with coercive fields significantly greater than the amplitude of the applied field for which the rate of magnetization reversal is significantly lower. In the fourth paragraph of the subsection “An ion channel gated by magnetic heating?”, it is stated that the organic shell surrounding ferritin would interfere with physical rotation, and this implicitly assumes particles with large magnetic anisotropy and high coercive field. Such an assumption could be made for the sake of argument, but should be made explicit.

4) The validity of bulk heat transport from a spherical object at the nanoscale requires some careful motivation. The result of applying this reasoning to synthetic MNPs that heat much more effectively than ferritin (larger, several hundred W/g) still yields a predicted temperature increase at the surface of nanoparticle of no more than 10-6-10-5 K. This simple model, however, fails to explain several recent experiments that indicate a temperature increase of several degrees K or even tens of degrees K that sharply declines within nanometers from MNP surfaces (e.g. Riedinger et al., Nano Lett. 2013; Yoo et al., Angew. Chem. Int. Ed. 2013; Dong and Zink, ACS Nano, 2014). While these works do not supply a satisfying theoretical explanation for these results, these experiments along with the wealth of studies from the field of nano-electronics indicate that phonon transport, rather than simple diffusion, must be considered to explain heat dissipation at these scales. However, even these models do not seem able to fully account for the magnitude of the discrepancy (G. Chen, 1996. J. Heat Transfer 118(3), 539-545). Due to the absence of a satisfactory physical model of heat transport at the MNP surface, the author is encouraged to discuss the heat dissipation from a perspective of existing experimental data. In the case of ferritin, the heating is implausible simply due to its negligible SLPs at the conditions reported by Stanley et al. and Wheeler et al.

5) Subsection “1. The protein complex has a permanent dipole moment”: Does the author really mean "blocking temperature" here? That is a characteristic of superparamagnetic particles and the temperature at which the "spins become locked to the molecular axes" sounds like an exchange interaction. So this is confusing.

6) Subsection “An ion channel gated by magnetic heating?”, third paragraph: Reorienting is accomplished by relaxation processes, not opposed by them. It would be better to say "opposed by magnetic anisotropy of the particles".

7) Subsection “An ion channel gated by magnetic heating?”, fifth paragraph: "Zero" is perhaps a bit extreme. It would be more precise to say that the heating rate is too low to be measurable.

8) Subsection “An ion channel gated by magnetic heating?”, fifth paragraph: "C" is a confusing choice for thermal conductivity because specific heat is typically represented as C. "k" or "kappa" might be better.

9) Subsection “Magnetic heating of nanoparticles”: "SLP varies proportionally to the frequency" This is not, strictly speaking, correct for superparamagnetic particles in either the framework of LRT or DHT. It should be made clear that this is an estimate perhaps reasonable for small adjustments and back of the envelope calculations, but not universally true."

---

## [Author Response]

*Essential revisions:*

*One of the reviewers suggested several modifications of wording, definitions, and equations in order that the commentary be technically accurate.*

In his manuscript “Physical limits to magnetogenetics” Prof. Markus Meister undergoes a series of back-of-the-envelope calculations that put into question physical mechanisms of magnetically- driven cell signaling proposed in a number of recent reports. First of all, I would like to applaud Prof. Meister for standing up for basic physics in light of recent articles that propose bio- magnetic mechanisms that are challenging to reconcile with the laws of magnetism. Given the recent interest in magnetoreception and magnetogenetics, I am confident that this is a timely article that should serve as a guide to the community. I do believe, however, that a number of minor improvements can be made to enhance the value of the manuscript. Below are a few suggestions for further clarification:

1) The manuscript would benefit from simple diagrams that illustrate forces and fields involved in all of the discussed scenarios. Simple figures would help our colleagues to develop conceptual understanding of magnetostatics.

Agreed. I have added Figure 1 to illustrate the various scenarios.

2) It would be clearer if all the equations were first spelled out symbolically prior to plugging in the numbers, as it would help our colleagues to use those equations in their own future calculations.

*For example: Eq. (1)*
μeff= n(n+2)μB= 5(5+2)μB
*where n = number of unpaired electrons.*

Made these adjustments.

3) Subsection “2. Individual complexes align with the earth's field”: “…the complex of 20 aligned Fe atoms…” should be “40 aligned Fe atoms”, to be consistent with the first sentence of the aforementioned subsection, as well as Eq. (3).

Corrected the typo.

4) Magnetization and magnetic moment should not be confused. M – is magnetization measured in A/m, magnetic moment m is measured in Am2 (=J/T – not a standard unit).

*5) Eq. (7) is not technically correct. It should be: M = X_v_H, where M is magnetization. In case of an isolated magnetic moment in air, in SI units one can write:*
M=Xv Bμo,
*where*
μo=4πx10−7[NA2= T.mA= JA2.m]
*is permeability of free space.*

6) The calculation of the susceptibility in the Methods section is rather confusing. When discussing the susceptibility, the author refers to volumetric susceptibility (X_v_ – a unitless quantity), while what he appears to mean is molar susceptibility (X_m_ in m^3^/mol (SI)). There are some typos in text and ambiguity in equations associated with this.

*Unit conversion for volumetric susceptibility is as follows:*
XSIv=4πXCGSv

*Unit conversion for molar susceptibility is:*
XSIm[m3mol]=4π x 10−6 XmCGS[cm3mol]
*(not vice versa as stated in Methods section.)*

If the author had indeed used volumetric (unitless) susceptibility the units would not work out and a factor of volume would emerge as expected for the distinction between magnetization and magnetic moment.

*Michaelis et al. measure ferritin molar susceptibility in CGS units as:*XmCGS=5.9 x  10−3[cm3mol]

*The susceptibility per Fe atom in SI units is then:*XSIFe= XmSINA= 4π x 10−6 XmCGSNA= 4π x 10−6 x 5.9 x10−3 6.02 x 1023 ≈1.2 x 10−31[m3atom]

Where N_A_ is the Avodagros number.

*If ferritin included 2400 Fe atoms, the moment per complex as a function of applied field B (paramagnetic material does not have an inherent moment in the absence of applied field) in free space would be:*mmic= NFe⋅XSIFe⋅Bμ0=2400 ⋅XSIFeμ0 B= XSIMicμ0 B

*In calculations of magnetic moment according to*
m= Xm Bμ0*, the author seems to inherently include dividing susceptibility by the permeability of free space into the definition of susceptibility, leading to unusual units:*XSIMicμ0= 2400x1.2x10−31[m3]4π x 10−7[JA2m]= 2.9x10−28[m3]4π x 10−7[JA2m]=2.3 x 10−22 [A2m4J=JA2m4J21T2=JT2]

Used different symbols to avoid confusion: *m* instead of *M* for magnetic moment, *ξ* instead of *χ* for the single-molecule magnetizability. Elaborated on the calculation of *ξ* from published versions of susceptibility.

7) Introduction, second paragraph: Feynman’s Lectures on Physics is an excellent physics reference, but the citation itself may require some editing. Profs. Leighton and Sands helped compile Feynman’s lectures into a written volume and are typically also credited with authorship, and since Feynman sadly passed away in 1988, a 2011 date without mention of the edition number may confuse some readers.

Cited earlier edition, along with URL.

8) Subsection “1. The protein complex has a permanent dipole moment”: “Those contain about 1 million iron atoms, closely packed to produce high exchange interaction, which serves to lock their individual magnetic moments in parallel (Feynman, 2011).”

This statement could use some clarification. Magnetite is ferrimagnetic, so the moments are not all parallel, and the ordering arises from superexchange mediated by oxygen. This does not subtract from point being made about scale and blocking temperature, but the author should carefully evaluate all instances of the term “ferromagnetic.”

Clarified ferrimagnetic.

*9) Subsection “A magneto-sensitive channel?”: “The hollow core of this particle can be filled with iron oxide (Arosio et al., 2009)” Upon examining this reference, it seems that ferritin may not store iron directly as iron oxide, but rather: “The main* in vitro *reaction of any ferritin type is to react with Fe(II) and induce its oxidation and deposition inside the cavity in a ferric oxyhydroxide core which is structurally similar to the mineral ferrihydrite.” (p. 590, Arosio et al.) Michaelis et al. (1943), as well as Papaefthymiou (2010) seem to generally agree on this point as well.*

Adjusted text.

10) The statement that “At room temperature, ferritin is strictly paramagnetic” seems to be supported by Michaelis et al. (1943) and other literature. However, the justification that follows is somewhat unclear. A “strict” paramagnet should not exhibit magnetic ordering at any scale without an applied field. In contrast, superparamagnetic particles can retain their local magnetic order above their blocking temperature, while the direction of magnetization undergoes rapid thermal fluctuation. A Langevin curve is observed for bulk measurements of such materials, but low field susceptibility is large (“super”) compared to strict paramagnets because the magnetic moments responding to the external field are coming from comparatively large, magnetically ordered clusters. As noted in the Methods section of this manuscript, consistent measurements have been performed on biologically derived ferritin that seem to indicate paramagnetism at room temperature. However, for perspective, it may be useful to direct readers to some work that discusses the possibility of ferrimagnetic ferrihydrite phases such as Michel et al. (2010). PNAS: 107 (7) 2787-2792 doi: 10.1073/pnas.0910170107

Adjusted text to “paramagnetic or superparamagnetic”. The important point is that ferritin has no permanent magnetic moment. I think citation of the review by Papaefthymiou is sufficient for background here.

*11) Eq. (8) does not express susceptibility in units that correspond to susceptibility. Rather, it is an estimate of the moment of a ferritin when multiplied by B. The following correction should be considered (see comment 6 above for the Methods section):*m= NFe⋅XSIFe⋅Bμ0= (2.4 x 10−22JT2)⋅B

Where N_Fe_ is the number of Fe per ferritin.

See response to points 4-6.

*12) Eq. (9) does not seem correct if M actually represents a magnetic moment*
m⇀*. See Feynman, Vol 2, equation 15.4. The energy of a magnetized dipole in a field is:*U= −m→⋅B→= − NFe⋅XSIFe⋅B2μ0

The justification for the factor of ½ should be clarified if retained.

*13) Suggested correction for Eq. (10):*F1= − ddx U=2 NFe⋅XSIFe⋅Bμ0⋅dBdx

The numerical estimate should come out to be the same, but the physical reasoning is clearer, at least to this reviewer. Nonstandard representations of susceptibility should be avoided if possible.

If *m* is a permanent magnetic moment, as in Feynman Vol II Eqn 15.4, then there is no factor of ½. But here *m* is an induced magnetic moment that is itself proportional to the field. That reduces the free energy to *U* = -½*mB*. For example, imagine pulling the magnetic particle out of the field gradient to infinity. At first it has a moment of *m*, but as the field weakens, the moment weakens as well and thus one needs less and less mechanical work to extract it from the field. A good discussion of this is in Jackson, Classical Electrodynamics, 3^rd^ edn, Ch 5.16, Eqn 5.150. I added this citation.

14) In Eq. (11), magnetization and moment are again confused. Simply substitute m for M and the same or similar numerical result should be obtained. Is the author using the same assumption of applied field B as in the previous example? Two paramagnetic bodies should not exhibit attraction without an applied field.

See response to points 4-6.

15) For Eq. (15), see the comment on Eq. (9), paying attention to sign. The energy of the particle in the field aligned with its easy axis should be negative, so that it is less than that of the particle with B applied along the hard axis. Otherwise the torque will be in an unphysical direction.

Corrected the sign.

*Another important point, but only strongly encouraged, not required:*

*"For the sake of completeness, it might also be a good idea to add a subsection on the physicality of the RF heating approach proposed by the Stanley et al. team in their 2015 paper and a more recent 2016 paper (Nature 531: 647-50). This is the one biophysical mechanism in the magnetogenetics papers that is not yet treated in the current manuscript. At the least, it would be worth commenting briefly on the validity of the RF heating approach in the Discussion section."*

I have to think that the reviewers knew what the outcome of this exercise would be. Following this suggestion I looked into the claims about magnetic heating and they are even more ridiculous than the ones regarding magnetic forces. I have now added a third section on this topic. Please check it for accuracy and readability.

[Editors' note: further revisions were requested prior to acceptance, as described below.]

*The manuscript has been improved but there are some remaining issues that need to be addressed before acceptance, as outlined below from Reviewer 3:*

*"The revised version of the manuscript ‘Physical limits to magnetogenetics’ addresses my comments to the earlier version. The added discussion of heating in magnetic nanomaterials, however, would benefit from substantial clarification, particularly in light of recent experimental observations of unexpected nanoscale heating effects. My specific suggestions are outlined below:*

*1) Magnetic nanoparticle (MNP) heating with low-radio frequency alternating magnetic fields has been an active area of research for 5 decades, and a number of comprehensive reviews are dedicated to the subject. Citing one of these reviews (e.g. Pankhurst, J. Phys. D Appl. Phys. 2003 or 2009) would be useful for the community. Most applications of MNP heating are in the field of cancer hyperthermia, and in fact these materials have been used in clinical trials to extend life span of glioblastoma patients by triggering tumor cell necrosis through bulk heating (Maier-Hauff, J. Neurooncol. 2011). Many of these nanomaterials are indeed based on synthetic magnetite, and its transition-metal-doped derivatives with dimensions in the range 6-60 nm. Citations to cancer hyperthermia reviews/ clinical trials should be mentioned in addition to those chosen by the author.*

Thanks for these pointers. Pankhurst 2003 is indeed a useful review for both magnetic pulling and heating. I also cite a review of cancer hyperthermia, along with the clinical article mentioned by the reviewer.

*2) Specific loss power (SLP) is indeed the main figure of merit used to compare MNP heating efficacy, however, the reference to Hergt should be updated because linear-response theory (LRT) has a limited range of validity. It approximates the Neél relaxation time as independent of the applied field magnitude, an approximation that is most valid for MNPs with coercive field significantly exceeding the amplitude of AMF. This is not the case for many MNPs in 10s kA/m fields. Recent articles by Carrey, Mamiya and colleagues (Carrey, J. Appl. Phys. 2011; Mamiya, Sci. Rep. 2011) offer a dynamic hysteresis theory (DHT) for heat dissipation by MNPs in AMFs. According to DHT, the particle loss per cycle is predicted to saturate at large fields rather than growing without bound as LRT suggests. For quantitative experimental evidence, see (Mehdaoui, Adv. Funct. Mater. 2011; Christiansen et al. Appl. Phys. Lett. 2014). Achieving non-negligible SLPs with small particles is sometimes possible by simply increasing the AMF frequency. This exact principle has been applied by Pralle and colleagues, who used AMF with an amplitude 2 kA/m and frequency 40 MHz to heat 6 nm MnFe2O4 MNPs and actuate TRPV1 (Huang, Nat. Nanotechnol. 2010). Prolonged operation at such frequencies, however, is not advisable due to the inductive heating of the biological tissue observed for field-frequency products in excess of 5×109 kA m-1 s^-1^ (Hergt and Dutz, J. Magn. Magn. Mater. 2007). Bearing this constraint in mind, larger particles with wider hysteresis loop areas are frequently chosen as more effective at reasonable field-frequency conditions.*

These are interesting developments, but I think an extended treatment of magnetic heating theory across particle sizes and experimental conditions is beyond the scope of the article. The focus is on ferritin and related particles under conditions similar to those of Stanley et al. 2015. The Hergt 2006 review combines a simple introduction to the theory with a good number of measurements relevant to this regime.

*3) When talking about heat dissipation, the author also mentions Brown relaxation in addition to Neél relaxation (described by DHT) in passing. Brown relaxation, i.e. physical rotation of the particles, is only applicable to MNPs with coercive fields significantly greater than the amplitude of the applied field for which the rate of magnetization reversal is significantly lower. In the fourth paragraph of the subsection “An ion channel gated by magnetic heating?”, it is stated that the organic shell surrounding ferritin would interfere with physical rotation, and this implicitly assumes particles with large magnetic anisotropy and high coercive field. Such an assumption could be made for the sake of argument, but should be made explicit.*

Removed this from the revised text.

*4) The validity of bulk heat transport from a spherical object at the nanoscale requires some careful motivation. The result of applying this reasoning to synthetic MNPs that heat much more effectively than ferritin (larger, several hundred W/g) still yields a predicted temperature increase at the surface of nanoparticle of no more than 10-6-10-5 K. This simple model, however, fails to explain several recent experiments that indicate a temperature increase of several degrees K or even tens of degrees K that sharply declines within nanometers from MNP surfaces (e.g. Riedinger et al., Nano Lett. 2013; Yoo et al., Angew. Chem. Int. Ed. 2013; Dong and Zink, ACS Nano, 2014). While these works do not supply a satisfying theoretical explanation for these results, these experiments along with the wealth of studies from the field of nano-electronics indicate that phonon transport, rather than simple diffusion, must be considered to explain heat dissipation at these scales. However, even these models do not seem able to fully account for the magnitude of the discrepancy (G. Chen, 1996. J. Heat Transfer 118(3), 539-545). Due to the absence of a satisfactory physical model of heat transport at the MNP surface, the author is encouraged to discuss the heat dissipation from a perspective of existing experimental data. In the case of ferritin, the heating is implausible simply due to its negligible SLPs at the conditions reported by Stanley et al. and Wheeler et al.*

I have included a short discussion of non-Fourier thermal transport and thermal contact resistance. As the reviewer states, and my numbers confirm, none of these extensions offer any explanation for the paradoxical claims of local heating. There is a good amount of skepticism in the literature about these claims, along with suggestions that the underlying thermometry methods should be reevaluated. The revised text cites this literature.

*5) Subsection “1. The protein complex has a permanent dipole moment”: Does the author really mean "blocking temperature" here? That is a characteristic of superparamagnetic particles and the temperature at which the "spins become locked to the molecular axes" sounds like an exchange interaction. So this is confusing.*

The confusing text has been replaced by "above which the magnetic moment fluctuates thermally".

*6) Subsection “An ion channel gated by magnetic heating?”, third paragraph: Reorienting is accomplished by relaxation processes, not opposed by them. It would be better to say "opposed by magnetic anisotropy of the particles".*

Done.

*7) Subsection “An ion channel gated by magnetic heating?”, fifth paragraph: "Zero" is perhaps a bit extreme. It would be more precise to say that the heating rate is too low to be measurable.*

Done.

*8) Subsection “An ion channel gated by magnetic heating?”, fifth paragraph: "C" is a confusing choice for thermal conductivity because specific heat is typically represented as C. "k" or "kappa" might be better.*

Done.

*9) Subsection “Magnetic heating of nanoparticles”: "SLP varies proportionally to the frequency" This is not, strictly speaking, correct for superparamagnetic particles in either the framework of LRT or DHT. It should be made clear that this is an estimate perhaps reasonable for small adjustments and back of the envelope calculations, but not universally true."*

Revised the text to state that this is an approximation. Recall that we are dealing with discrepancies of 9 log units, so a small frequency-dependence of susceptibility hardly matters for those arguments.